# RGRL: Quantum State Control via Representation-Guided Reinforcement Learning

## Abstract

Accurate control of quantum states is crucial for quantum computing and other quantum technologies. In the basic scenario, the task is to steer a quantum system towards a target state through a sequence of control operations. Determining the appropriate operations, however, generally requires information about the initial state of the system. Gathering this information becomes increasingly challenging when the initial state is not *a priori* known and the system's size grows large. To address this problem, we develop a machine-learning algorithm that uses a small amount of measurement data to construct its internal representation of the system's state. The algorithm compares this data-driven representation with a representation of the target state, and uses reinforcement learning to output the appropriate control operations. We illustrate the effectiveness of the algorithm showing that it achieves accurate control of unknown many-body quantum states and non-Gaussian continuous-variable states using data from a limited set of quantum measurements.

## 1 Introduction

Controlling the state of noisy intermediate-scale quantum systems Preskill (2018) is a major challenge for quantum computing and other quantum technologies. In recent years, reinforcement learning (RL) François-Lavet et al. (2018) has emerged as a powerful strategy for designing quantum control policies Chen et al. (2013a); Bukov et al. (2018); Bukov (2018); Zhang et al. (2019); Yao et al. (2021); Borah et al. (2021); Guo et al. (2021); Sivak et al. (2022); Porotti et al. (2022); Metz & Bukov (2023); Reuer et al. (2023). One of the benefits of this approach is that, unlike conventional model-based quantum control, RL can be used to adaptively learn quantum control policies without any knowledge of the underlying quantum dynamics Sivak et al. (2022). Beyond quantum control, RL has found applications in quantum information science, including quantum error correction Fösel et al. (2018); Nautrup et al. (2019); Zeng et al. (2023), quantum simulation Bolens & Heyl (2021), quantum compilation Zhang et al. (2020); Fösel et al. (2021); Moro et al. (2021), quantum sensing Xiao et al. (2022) and quantum communications Wallnöfer et al. (2020).

Many of the existing control methods focus on the preparation of a known target state from a known initial state Chen et al. (2013a); Bukov et al. (2018); Zhang et al. (2019); Porotti et al. (2022); Sivak et al. (2022); Metz & Bukov (2023). In this task, RL algorithms are often used to learn control policies from exact state descriptions. In practice, however, limited device calibration and imperfections in the setup can lead to uncertainty on the initial state of the system. It is then important to supplement RL algorithms with a state characterization step, in which useful information is gathered from quantum measurements. In many of the existing protocols, the relevant piece of information is the fidelity between the system's state and the target state Chen et al. (2013a); Bukov et al. (2018); Bukov (2018); Zhang et al. (2019); Porotti et al. (2022); Metz & Bukov (2023). Estimating the fidelity through measurements, however, becomes challenging as the system size grows Flammia & Liu (2011); da Silva et al. (2011). To circumvent this problem, a few works explored the possibility of directly using measurement outcomes for reward calculation Reuer et al. (2023); Borah et al. (2021); Sivak et al. (2022), but scalability still remains a challenge.

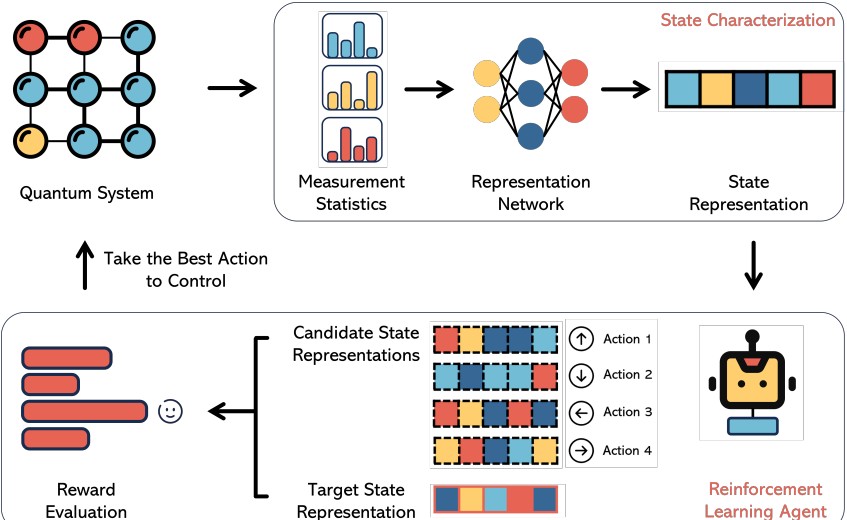

Figure 1: Schematic of our algorithm for quantum state characterization and control. The measurement statistics obtained by probing a quantum system are fed into a representation network, which builds its own representation of the quantum state. Using the state representation as input, a reinforcement learning agent evaluates the candidate state representations after performing possible control actions. The agent then selects the optimal action to maximize the expected reward, aiming to minimize the distance between the controlled state representation and the target state representation.

In this paper, we introduce a **R**epresentation-**G**uided **R**einforcement **L**earning (**RGRL**) algorithm, which combines a neural network for learning unknown quantum states with an RL algorithm to steer an uncharacterized quantum state towards a target state using only measurement statistics. The core of our algorithm is a representation network Zhu et al. (2022) that produces its internal representation of quantum states and uses it to estimate a measure of the similarity between the state under control and the target state Wu et al. (2023a). The similarity estimate is then used in the RL stage to compute rewards and to determine the control operations needed to steer the quantum system to the desired state. Overall, our algorithm provides an illustration of how a machine without any built-in knowledge of quantum physics can learn to control a piece of quantum hardware. The effectiveness of this approach is illustrated through numerical simulations, showing strong performance for various scenarios of quantum control. The contributions are:

(1) We develop a novel algorithm, RGRL, that combines state representations with reinforcement learning for quantum control. By leveraging neural representations constructed from limited measurement data, the RGRL algorithm can evaluate the similarity between the current and target states and determine the optimal control actions. This approach facilitates adaptive learning of control policies without requiring pre-existing knowledge of the quantum system's dynamics.

(2) We provide a scalable solution for quantum state preparation with limited initial state information. One of the major challenges in quantum state control is dealing with the uncertainty of the initial state, especially as the system size increases. Our RGRL algorithm addresses this issue by using measurement statistics to construct an internal representation of the quantum state. This approach significantly reduces the reliance on precise initial state information, making the algorithm scalable and applicable to larger quantum systems.

(3) We demonstrate the algorithm's effectiveness in controlling many-body ground states and continuous-variable states. Specifically, we demonstrate its application to controlling many-body ground states, which are crucial for understanding complex quantum systems and phase transitions. Additionally, we apply the algorithm to continuous-variable states, which are essential for quantum error correction and other quantum information processes. These numerical results highlight the algorithm's robustness and adaptability across different types of quantum states and systems, validating its broad utility in practical quantum technologies.

## 2 PRELIMINARIES: QUANTUM STATE AND MEASUREMENT

Here we briefly present the preliminary information necessary for understanding the results in our paper.

**Quantum state and quantum measurement.** A quantum state is a mathematical representation of a physical system encapsulating all the information about that system. A quantum state, mathematically, can be represented by a vector in a Hilbert space. Given a $d$-dimensional Hilbert space with computational basis $\{|i\rangle\}_{i=1}^d$, a pure state $|\phi\rangle = \sum_{i=1}^d \alpha_i |i\rangle$ is a superposition of states, where $\alpha_i \in \mathbb{C}$ satisfy $\sum_{i=1}^d |\alpha_i|^2 = 1$. A mixed state is a convex combination of pure states.

We use the notation $\mathbf{M} = (\mathbf{M}_i)_{i=1}^n$ to denote a quantum measurement, described by an $n$-outcome positive operator-valued measure (POVM) that associates the measurement outcome $i$ with a positive operator $\mathbf{M}_i$, satisfying the normalization condition $\sum_{i=1}^n \mathbf{M}_i = \mathbf{I}$ ($\mathbf{I}$ is the identity operator on the Hilbert space). For qubit systems, the most common quantum measurement is single-qubit Pauli measurement, i.e. projective measurement on the eigenbasis of either one of the Pauli matrices $\sigma^x := \begin{pmatrix} 0 & 1 \\ 1 & 0 \end{pmatrix}$, $\sigma^y := \begin{pmatrix} 0 & -\mathrm{i} \\ \mathrm{i} & 0 \end{pmatrix}$, and $\sigma^z := \begin{pmatrix} 1 & 0 \\ 0 & 1 \end{pmatrix}$. For each single-qubit Pauli measurement, there are only two possible outcomes, corresponding to states $|+\rangle := \frac{1}{\sqrt{2}}(|0\rangle + |1\rangle)$ and $|-\rangle := \frac{1}{\sqrt{2}}(|0\rangle - |1\rangle)$ after measuring $\sigma_x$, $|+\mathrm{i}\rangle := \frac{1}{\sqrt{2}}(|0\rangle + \mathrm{i}|1\rangle)$ and $|-\mathrm{i}\rangle := \frac{1}{\sqrt{2}}(|0\rangle - \mathrm{i}|1\rangle)$ after measuring $\sigma_y$, and $|0\rangle$ and $|1\rangle$ after measuring $\sigma_z$. An $n$-qubit Pauli measurement is measuring each single qubit in a Pauli basis independently and has overall $3^n$ options.

**Many-body ground state.** Here we briefly introduce the many-body ground states under investigation in this paper. Given a Hamiltonian, its ground states are defined as the eigenstates of the Hamiltonian with lowest energy. In this paper, we consider the following Hamiltonian, known as bond-alternating XXZ model Elben et al. (2020),

$$H_{\text{XXZ}} = J \sum_{i=1} \left( \sigma_{2i-1}^x \sigma_{2i}^x + \sigma_{2i-1}^y \sigma_{2i}^y + \delta \sigma_{2i-1}^z \sigma_{2i}^z \right) + J' \sum_{i=1} \left( \sigma_{2i}^x \sigma_{2i+1}^x + \sigma_{2i}^y \sigma_{2i+1}^y + \delta \sigma_{2i}^z \sigma_{2i+1}^z \right),$$

where $J, J', \delta \in \mathbb{R}$ are physical parameters. Here, each subscript denotes the site of qubit, and the tensor product notations between Pauli matrices on different sites are omitted. In our numerical experiments, we perform Pauli measurements on three-qubit subsystems, measuring each of the three neighbouring qubits independently on a Pauli basis while tracing out all the other qubits.

The ground state space of a Hamiltonian can include different phases of matter. The quantum states within the same quantum phase exhibit similar physical property, whereas the physical behaviours of those states across the boundaries between two quantum phases become exotic, which is known as quantum phase transition. Specifically, $H_{\text{XXZ}}$ is parametrized by two independent parameters $J/J'$ and $\delta$. In the parameter space $(J/J', \delta) \in (0, 3) \times (0, 4)$, there are three phases of matter: the topological symmetry-protected topological (SPT) phase, the trivial SPT phase, and the symmetry-broken phase, as indicated by different colors in Fig. 3**b**.

**Optical continuous-variable state.** Rather than falling within a finite-dimensional Hilbert space, continuous-variable (CV) states Serafini (2017) fall on an infinite-dimensional Hilbert space, spanned by number states $\{|n\rangle\}_{n=0}^\infty$. A coherent state $|\alpha\rangle := \mathrm{e}^{-\frac{|\alpha|^2}{2}} \sum_{n=0}^\infty \frac{\alpha^n}{\sqrt{n!}} |n\rangle$ is characterized by its physical parameter $\alpha \in \mathbb{C}$ and can be obtained by applying a displacement operation $D(\alpha)$ on the vacuum state $|0\rangle$, i.e. $|\alpha\rangle = D(\alpha) |0\rangle$. For data generation, we truncate the Hilbert space to a finite value $N_{\max}$ and simulate all the states within the truncated space $\mathrm{span}\left(\{|n\rangle\}_{n=0}^{N_{\max}}\right)$.
A particular CV quantum gate we apply on coherent state inputs is the Kerr gate, driving any simple Gaussian state to a complex nonGaussian state. A common quantum measurement in CV quantum systems is homodyne measurement characterized by physical parameter $\theta \in [0, \pi)$, i.e. projective measurements on the eigenbasis of $\cos\theta \hat{x} + \sin\theta \hat{p}$ with $\hat{x}$ and $\hat{p}$ denoting position operator and momentum operator respectively. The outcome of each homodyne measurement is a bounded real number in our classical simulation. The collected outcome statistics of a homodyne measurement is the projection of the Wigner function of a CV state along that direction.

## 3 THE RGRL ALGORITHM

### 3.1 OVERVIEW

Our algorithm applies to the scenario where a classical learning agent gathers information from and performs actions on a quantum system. Following RL terminology, we will refer to the quantum system as the agent's environment. The interactions between the agent and the environment are mediated by an experimenter, who performs quantum measurements and other quantum operations based on the agent's instructions. Our algorithm consists of two stages: quantum state characterization/learning and quantum control, as depicted in Fig. 1. In the state learning stage, the agent has access to many copies of an unknown quantum state. The set of quantum measurements that can be performed in the state is denoted by $\mathcal{M}$. At the beginning of the protocol, the experimenter randomly chooses a subset $\mathcal{S} \subset \mathcal{M}$ of quantum measurements. In each control step, the experimenter performs each measurement $\mathbf{M} \in \mathcal{S}$ in multiple copies of the state $\rho$, obtaining the frequency distribution of the outcome $\boldsymbol{d}_M := \mathrm{Tr}(\rho\mathbf{M})$. The set of measurement statistics $\mathcal{D} := \{\boldsymbol{d}_M\}_{\mathbf{M} \in \mathcal{S}}$ is fed into a (pre-trained) representation network to produce the data-driven state representation $\boldsymbol{r}_\rho$. The distance between $\boldsymbol{r}_\rho$ and $\boldsymbol{r}_{\rho_t}$ in the representation space is taken as an indicator of the distance between $\rho$ and $\rho_t$, where $\boldsymbol{r}_{\rho_t}$, i.e., the representation of the target state $\rho_t$, can be obtained from the simulated measurement data of $\rho_t$ corresponding to the measurement set $\mathcal{S}$.

In the $k$th control step, the agent associated with the RGRL algorithm receives the measurement data $\mathcal{D}_k$ from the environment as its partial observation and selects an action $a_k \in \mathcal{A}$ ( from a finite set of predefined actions based on the policy $\pi$ and the measurement observations $\mathcal{D}_k$. Each action corresponds to a tuning of the physical parameters. Upon receiving the action determined by the RGRL algorithm, the experimenter implements it on the system, driving the current quantum state to a new state in the next control step. This process is repeated in each environment-agent interaction cycle.

Here, the control policy is modeled by a conditional probability $\pi_{\boldsymbol{\theta}}(a|\mathcal{D})$ of action $a$ depending on partial observations $\mathcal{D}$, parameterized by a neural network with parameters $\boldsymbol{\theta}$. At the $k$th step, a reward $r_k \propto -||\boldsymbol{r}_{\rho_k} - \boldsymbol{r}_{\rho_t}||$ is assigned, where $\rho_k$ denotes the quantum state at the $k$th step, and $|| \cdot ||$ denotes the Euclidean distance in the representation space. Note that the partial observation $\mathcal{D}$ can be mapped to a dense representation vector $\boldsymbol{r}$ with a pretrained neural network. From the initial state to a terminal state, one full trajectory of tuples $\{(\mathcal{D}_k, a_k, r_k)\}_{k=1}^T$ is called an episode, where $T$ denotes the maximum length of an episode. For each episode, a cumulative reward $R := -\sum_{k=1}^T \gamma^{k-1} r_k$ is assigned, where $\gamma$ denotes the discount rate. The goal of the RL algorithm is to learn a policy $\pi_{\boldsymbol{\theta}} : \mathcal{D} \to \mathcal{A}$ that maximizes the averaged cumulative reward $\bar{R}$ over multiple episodes by optimizing the parameters $\boldsymbol{\theta}$ through gradient ascent, i.e., $\Delta\boldsymbol{\theta} \sim \nabla_{\boldsymbol{\theta}}\bar{R}$ Sutton et al. (1999); Silver et al. (2014). More details about this neural network structure can be found in the supplementary material. This model-free RL algorithm can discover the optimal policy without knowledge of the explicit model of the quantum dynamics of the environment.

### 3.2 STRUCTURE AND PRETRAINING OF REPRESENTATION NETWORK

Here, we introduce the implementation of the representation network utilized in our framework. As illustrated in Figure 2a, the representation network $f_{\boldsymbol{\xi}}$ takes as input the parameterization $\boldsymbol{m}_i$ of the measurement $M_i \in \mathcal{S}$ and its outcome statistics $\boldsymbol{d}_i$ for the specific state $\rho$. For each pair $\mathcal{D}_i = (\boldsymbol{m}_i, \boldsymbol{d}_i)$, the representation network produces a vector $\boldsymbol{r}_i = f_{\boldsymbol{\xi}}(\boldsymbol{m}_i, \boldsymbol{d}_i)$. These vectors, corresponding to different pairs, are then combined into a single vector $\boldsymbol{r}$ by an average function.

We utilize two distinct methods to train the representation network. When only measurement data of the quantum states is available, we employ the GQNQ architecture proposed by Zhu et al. in Zhu et al. (2022). Specifically, we conduct self-supervised learning based on these measurement data. The process involves inputting the measurement data into the representation network to generate representations of the quantum states. Subsequently, we reconstruct the measurement statistics from these representations using a generation network shown in Figure 2b. During training, we minimize the reconstruction loss between the predicted measurement statistics $\boldsymbol{p}_i$ and the ground truth $\boldsymbol{d}_i$ for each measurement $M_i \in \mathcal{S}$ .

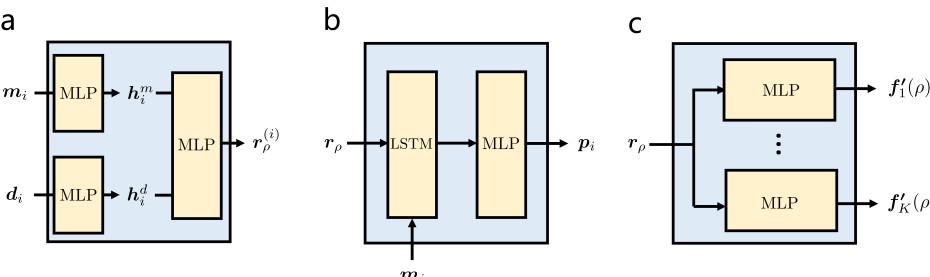

Figure 2: Structure and pretraining of the representation network. Subfigure **a** displays an abstract structure of the representation network. Subfigures **b** and **c** present the decoder network used for unsupervised and supervised pretraining of the representation network, respectively.

When properties of the quantum states, such as mutual information between subsystems, are available, we utilize the architecture introduced in Wu et al. (2023b). In this approach, the representations generated by the representation network are fed into a prediction network shown in Figure 2c. During training, given the availability of $K$ labeled property values $\{f_1(\rho), \cdots, f_K(\rho)\}$, we employ a supervised learning manner to minimize the loss between the estimated and true values of these properties. This ensures that the combination of the learned representations and the prediction network can accurately predict the $K$ properties.

### 3.3 REINFORCEMENT LEARNING ALGORITHMS

**Environment.** The environment for the RGRL algorithm is modeled as a quantum oracle, which applies the control policy to the quantum system and retrieves measurement data from the controlled quantum state. Based on the control policy generated by the RGRL algorithm, the oracle adjusts the physical parameters accordingly. The measurement data consist of statistical outcomes obtained from quantum states measured on various bases.

**Algorithm.** The environment of controlling the ground state can be modeled as a Markov decision process (MDP), therefore, it is highly suitable for RL Sutton & Barto (2018). We make use of model-free RL as the intelligent agent to optimize the control policies. The input of the RL policy network denoted as $r$, is the neural representation of the current ground quantum state. The action $a$ controls the tuning of the physical parameters. The reward is proportional to the negative distance between the current quantum state and the target quantum state in the representation space. More precisely, the $k$th control step of the reward is given by

$$r_k = -\frac{\|\boldsymbol{r}_{\rho_k} - \boldsymbol{r}_{\rho_t}\|_2}{\sqrt{d}}, \tag{1}$$

where $d$ is the dimension of the representation space. The goal of RL is to find an optimal policy for the agent to obtain optimal rewards. We use policy gradient methods, which aim at modeling and optimizing the policy directly. Specifically, the PPO method is applied to build the RGRL algorithm Schulman et al. (2017). The detailed mathematical descriptions and pseudocode can be found in Appendix B.

## 4 EXPERIMENTS

### 4.1 CONTROL OF PHASE TRANSITION IN MANY-BODY SYSTEMS

We first apply our RL algorithm for control of intermediate-scale many-body ground state across phase transition, which is of great significance in the field of many-body physics Pollmann et al. (2012); Chen et al. (2013b); Smith et al. (2022). Specifically, we consider the ground states of a 50-qubit bond-alternating XXZ model. We discretize the parameter space $(J/J', \delta) \in (0, 3) \times (0, 4)$ into a $21 \times 21$ grid. Assuming both the initial state and the target state fall within this grid but in different phases, we proceed as follows at each control step.

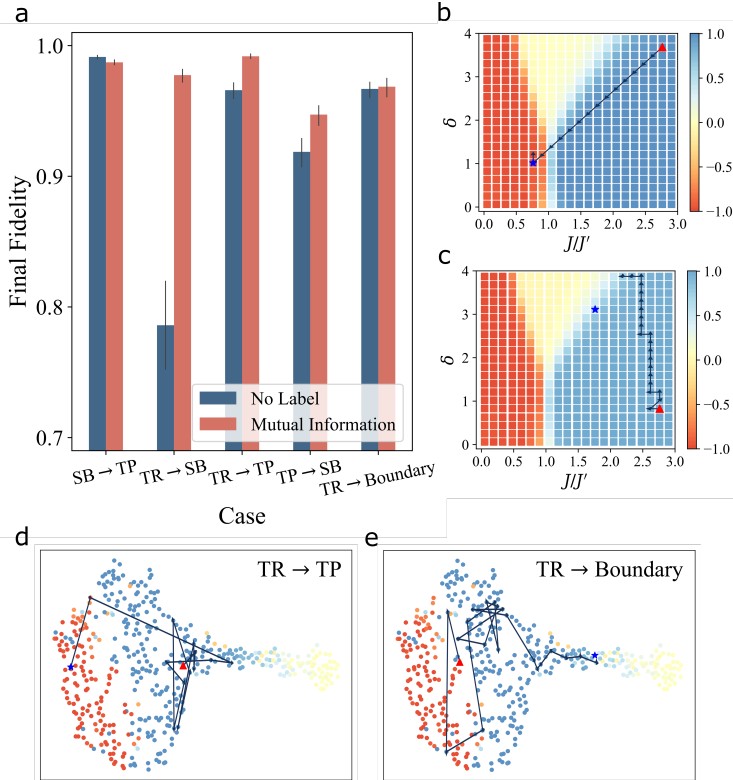

Figure 3: Control of many-body ground states across phase transition in the bond-alternating XXZ model. Subfigure **a** displays the quantum fidelities between the controlled quantum state and the target state in five different control scenarios after 30 control steps, averaged over 500 experiments, where SB denotes symmetry broken phase, TP dentoes topological SPT phase and TR denotes trivial SPT phase. The vertical lines atop each bar denote the 95% confidence intervals. Subfigures **b** and **c** illustrate examples of the quantum evolution trajectory of the controlled state from the trivial SPT phase to the topological SPT phase and from the trivial SPT phase to the phase boundary in the phase diagram, respectively. In these phase diagrams, the red triangle represents an initial ground state, the blue star represents a target ground state, and the red, yellow, and blue areas denote the topological SPT, symmetry broken, and trivial SPT phases, respectively. Subfigures **d** and **e** depict the trajectories of the controlled state corresponding to the trajectories in Subfigures **b** and **c**, respectively, in the representation space, projected by the t-SNE algorithm.

Instead of measuring every single qubit in the spin chain, we perform single-qubit Pauli measurements only on neighboring three-qubit subsystems, which correspond to a marginal of the entire system. We select 50 different measurements, each corresponding to a Pauli string on three nearest-neighbor qubits, and use this same set of measurements at each control step. After performing the quantum measurements, we feed the measurement statistics of short-range spin correlations into the neural network. The RGRL algorithm then outputs an action to tune the pair of parameters $(J/J', \delta)$, chosen from the set of eight options: $(J/J', \delta) \leftarrow \{(J/J' \pm 0.15, \delta), (J/J', \delta \pm 0.2), (J/J' \pm 0.15, \delta \pm 0.2)\}$.

We investigate five different control scenarios, each corresponding to a different pair of initial and target states: (1) Control a ground state in the symmetry-broken phase towards the topological SPT phase. (2) Control a ground state in the trivial SPT phase towards the symmetry-broken phase. (3) Control a ground state in the trivial SPT phase towards the topological SPT phase. (4) Control a ground state in the topological SPT phase towards the symmetry-broken phase. (5) Control a ground state in the trivial SPT phase towards the phase boundary between the trivial SPT and symmetry-broken phases. Figure 3 shows the quantum fidelity between the controlled state and the target state for these five scenarios after 30 control steps.

To demonstrate that the quality of state representations affects the performance of quantum control, we investigate the same control scenarios using two different types of state representations. The first type is obtained by predicting the outcome statistics of measurements that have not yet been performed, as described in Ref.Zhu et al. (2022). The second type is obtained for predicting the order-two Rényi mutual information between two subsystems of a quantum state, as outlined in Ref. Wu et al. (2023b). Our results in Figure. 3**a** indicate that using state representations designed for predicting Rényi mutual information leads to higher quantum fidelity in our control scenarios. This improvement is attributed to the fact that these state representations capture the nonlinear properties of quantum states, thereby more clearly distinguishing different topological phases of matter.

Figure 3**b** shows an example of the trajectory of the ground state evolution under control, transitioning from a trivial SPT phase to a topological SPT phase. Figure 3**c** presents an example of a controlled ground state moving from a trivial SPT phase towards the phase boundary. Our RGRL algorithm finds the optimal path in phase space for controlling a ground state from the trivial SPT phase to the topological SPT phase, as illustrated in Figure 3**b**. For controlling a ground state in the trivial SPT phase towards a state across the phase boundary, the controlled state successfully reaches the phase boundary but fails to arrive at the exact critical point, as shown by Figure 3**c**. This is because the state representations obtained by our representation network are quite similar near the phase boundary, preventing the RL algorithm from accurately distinguishing different states across the boundary.

Figures 3**d** and **e** illustrate the control trajectory in state representation space, corresponding to the ground state evolution trajectories in Figures 3**b** and **c**, projected onto a 2D plane using the t-SNE algorithm. By projecting the state representations of all ground states on the grid onto a 2D plane, we find that in representation space, the state representations do not follow their pattern in phase space. Thus, the optimal path in phase space does not correspond to the shortest trajectory in the representation space, implying that the control task we consider is highly nontrivial.

We used the state representations for predicting mutual information to plot the control trajectories in Figures 3. We also compare the quantum control trajectories using the state representations for predicting measurement statistics (Figure 4) with those based on state representations to predict mutual information (Figure 5). The results indicate that the former takes fewer control steps to arrive at the target state than the latter, demonstrating that higher-quality state representations yield higher control efficiency. To demonstrate this, we present the trajectories of the controlled states in the representation space in Figure 6. It can be observed that the positions of state representations are more separate when mutual information is used, enabling the algorithm to construct the correct trajectory with fewer steps.

## 4.2 GENERATING THE OUTPUT OF A QUANTUM PROCESS

As an additional application, we employ our RGRL algorithm to produce a target output from an unknown quantum process. We consider the scenario where a known target output state can be obtained by applying the unknown quantum process to a certain input state. The goal is to find an input that causes the quantum process to approximately generate the target output.

To achieve this, we apply the unknown quantum process to multiple copies of a randomly chosen input state, resulting in multiple copies of the corresponding output state. By performing randomized measurements on the output, we estimate the similarity between the actual output state and the target output state. Based on the estimated similarity, we then modify the input preparation. This learning-and-control cycle is repeated until we identify an appropriate input that produces an accurate approximation of the target output.

As an example, we apply our algorithm to prepare target output states of a CV Kerr quantum gate. The input states can be selected from the set of coherent states $|\alpha\rangle$, where $\alpha = re^{i\psi}$, $r \in [0, 3]$ and $\psi \in [0, 2\pi)$. Three homodyne measurement bases are randomly chosen, constituting the set of measurements $\mathcal{S}$ to be performed on the output at each control step. Taking the measurement statistics corresponding to the measurements in $\mathcal{S}$ as input, our RGRL algorithm determines the tuning of the parameter $\alpha$ out of four options: $(|\alpha|, \arg(\alpha)) \leftarrow \{(|\alpha| \pm 0.09, \arg(\alpha) \pm 0.06\pi)\}$. The measurement-and-control cycle is depicted in Fig. 7**a**. Figure 7**b** illustrates the quantum fidelity between the generated output state and the target output state within 55 control steps, averaged over

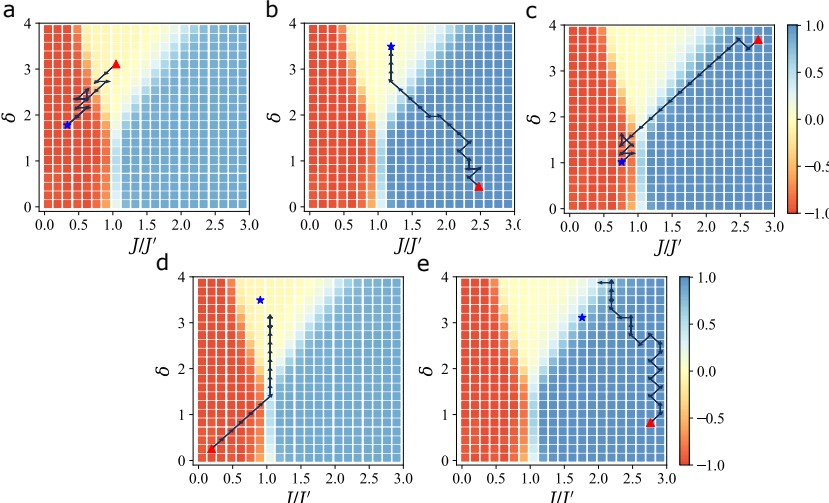

Figure 4: Trajectory of the controlled state, utilizing state representations to predict measurement statistics: **a** from symmetry broken phase to topological SPT Phase, **b** from trivial SPT phase to symmetry broken phase, **c** from trivial SPT phase to topological SPT Phase, **d** from topological SPT Phase to symmetry broken phase, **e** from the trivial SPT phase to the phase boundary in the phase diagram.

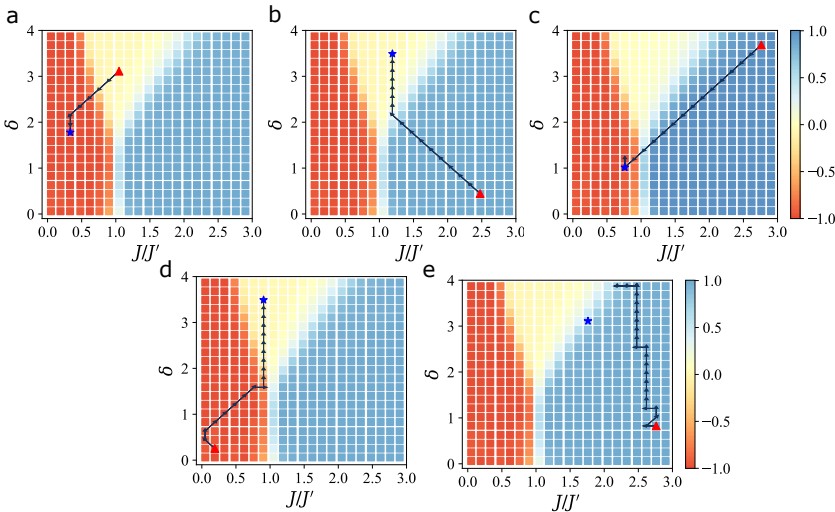

Figure 5: Trajectory of the controlled state, utilizing state representations to predict mutual information: **a** from symmetry broken phase to topological SPT Phase, **b** from trivial SPT phase to symmetry broken phase, **c** from trivial SPT phase to topological SPT Phase, **d** from topological SPT Phase to symmetry broken phase, **e** from the trivial SPT phase to the phase boundary in the phase diagram.

200 random pairs of initial input states and target output states. The results indicate that, although the initial output state fidelity is nearly zero, the quantum fidelity exceeds 0.9 after 50 control steps.

## 5 CONCLUSIONS

We developed a reinforcement-learning algorithm for steering a quantum system, initially in an unknown state, to a given target state. In our algorithm, each control action is determined by a neural network based on measurement data from a small set of quantum measurements. The

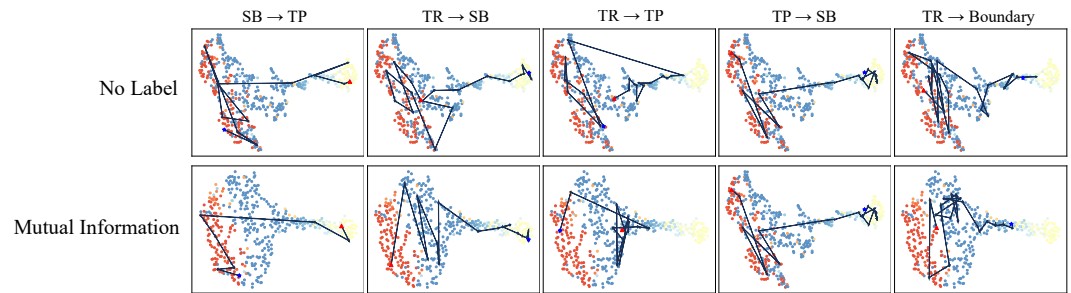

Figure 6: Trajectories of the controlled state under control in the representation spaces.

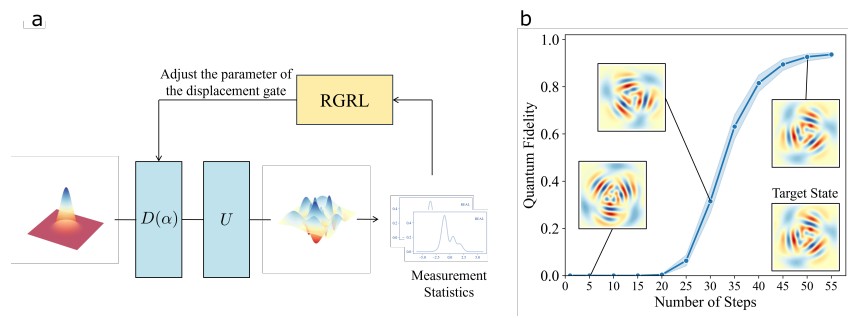

Figure 7: Producing target output of CV unitary quantum process. Subfigure **a** illustrates the measurement-and-control loop for producing the target output of a quantum process $\mathcal{U}$ by controlling the parameter $\alpha$ of a displacement gate applied to a vacuum state. Subfigure **b** shows the quantum fidelity between the output and the target within 55 control steps, averaged over 200 randomly chosen pairs of initial and target states, along with several examples of Wigner function Serafini (2017) of the controlled states.

measurement set is randomly chosen, independently of the target quantum state, thereby offering flexibility for a broad set of applications involving the control of uncharacterized, intermediate-scale quantum systems using a limited set of quantum measurements. Our method provides a novel way to learn and control the quantum state, which also highlights the significance of representation learning and RL in quantum computing and information processing.

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

## A  RELATED WORKS

Numerous studies have explored the use of neural networks to learn quantum systems Carleo & Troyer (2017); Torlai et al. (2018); Carrasquilla et al. (2019); Zhu et al. (2022); Schmale et al. (2022) and predict their properties Zhang et al. (2021); Xiao et al. (2022); Du et al. (2023); Wu et al. (2023a); Qian et al. (2024); Gao et al. (2018); Gray et al. (2018); Koutný et al. (2023); Torlai et al. (2018; 2019; 2018; 2019); Carrasquilla et al. (2019); Smith et al. (2021); Schmale et al. (2022); Tang et al. (a;b). Along this line, an interesting approach involves learning concise state representations Zhu et al. (2022) from the outcome data of randomized measurements Huang et al. (2020); Elben et al. (2023). These data-driven representations preserve essential information about quantum states and enable the assessment of their similarities Wu et al. (2023a). This work advances the field by integrating reinforcement learning (RL) to develop a representation-guided RL algorithm for efficient quantum control.

While many existing RL-based quantum control methods Chen et al. (2013a); Bukov et al. (2018); Zhang et al. (2019); Porotti et al. (2022); Sivak et al. (2022); Metz & Bukov (2023) focus on preparing a known target state from a specified initial state, our algorithm runs in a state-agnostic manner. The only input to the neural network is the outcome data from randomized measurements, without any description of the state being controlled, including the initial state. This is a key distinction from previous approaches. Many quantum control strategies have used quantum fidelity between states for reward calculations Chen et al. (2013a); Bukov et al. (2018); Bukov (2018); Zhang et al. (2019); Porotti et al. (2022); Metz & Bukov (2023). However, in our scenario, quantum fidelity is

not accessible. Instead, we use the distance between state representations as a proxy for quantum fidelity when calculating rewards.

## B  RL ALGORITHMS

Note that for following the convention notation in RL, we replace the neural representation $r$ in the main text to $s$. In addition, in fact the observation of the RGRL algorithm is $\mathcal{D}$, i.e. the measurement statistics, we always map it to $r$ with a pretrained and frozen neural network. Therefore, we present the policy gradient directly with $r$ (here repalced with $s$) rather than $\mathcal{D}$. When considering the policy-gradient method, the expected reward function

$$J(\theta) = \sum_{s\in\mathcal{S}} d^\pi(s) \sum_{a\in\mathcal{A}} \pi_\theta(a|s) Q^\pi(s,a), \tag{2}$$

where $d^\pi(s)$ is the stationary distribution of Markov chain for policy $\pi_\theta$, $Q^\pi(s,a)$ is the value function when given by a policy $\pi$. $\mathcal{A}, \mathcal{S}$ denote the action space and observation spaces, respectively. Here, we omit notation $\theta$ when the policy $\pi_\theta$ is present in the subscript of other functions. The policy gradient theorem Sutton et al. (1999) states that the gradient over the reward function is given by

$$\nabla_\theta J(\theta) \propto \sum_{s\in\mathcal{S}} d^\pi(s) \sum_{a\in\mathcal{A}} Q^\pi(s,a) \nabla_\theta \pi_\theta(a|s),$$
$$= \mathbb{E}_\pi \left[ Q^\pi(s,a) \nabla_\theta \ln \pi_\theta(a|s) \right], \tag{3}$$

where $\mathbb{E}_\pi$ refers to $\mathbb{E}_{s\sim d^\pi, a\sim\pi_\theta}$ when both state and action distributions follow the policy $\pi_\theta$. Eq. (3) lays the foundation of policy gradient algorithms in RL. Meanwhile, Eq. (3) has no bias but have large variance. Many methods focus on reducing the variance of the estimated gradient while keeping the bias unchanged. Therefore, during updating the policy parameters, we often use the advantage function $A(s,a) = Q(s,a) - V(s)$ rather than the $Q$ function. $V(s)$ is the state value function used to evaluate the expected reward of current state whatever actions it takes. To better estimate the state value function, we often use Actor-Critic architecture to model the policy gradient algorithm. Critic network is used to estimate the state value function and actor network is used to model the policies.

In order to improve the sample efficiency and exploration ability, off-policy gradient methods are often employed. More formally, suppose the generated data trajectories obey the behavior policy $\beta(a|s)$, the objective function sums up the reward over the state distribution defined by this behavior policy

$$J(\theta) = \sum_{s\in\mathcal{S}} d^\beta(s) \sum_{a\in\mathcal{A}} Q^\pi(s,a) \pi_\theta(a \mid s),$$
$$= \mathbb{E}_{s\sim d^\beta} \left[ \sum_{a\in\mathcal{A}} Q^\pi(s,a) \pi_\theta(a \mid s) \right], \tag{4}$$

where where $d^\beta$ is the stationary distribution of the behavior policy $\beta$. Note that $\pi$ refers to the target policy which is used to estimate the state-action function, i.e. $Q(s,a)$. Subsequently, we can rewrite Eq. (3) as follows

$$\nabla_\theta J(\theta) = \mathbb{E}_\beta \left[ \frac{\pi_\theta(a|s)}{\beta(a|s)} Q^\pi(s,a) \nabla_\theta \ln \pi_\theta(a|s) \right], \tag{5}$$

where we call $\frac{\pi_\theta(a|s)}{\beta(a|s)}$ as importance weight. The off-policy gradient Eq. (5) implies that we can use behaviour policy generated trajectories to update the policy parameters. One important fact is that we omit the term of gradient over $Q$ function i.e. $\nabla_\theta Q^\pi(s,a)$. Fortunately, it turns out that approximated gradient with the gradient of $Q$ ignored still guarantees the policy improvement and eventually achieve the true local minimum Degris et al. (2012).

Trust region policy optimization (TRPO) algorithm Schulman et al. (2015) is used to stabilize the training process, namely avoiding parameter updates that change the policy too much at one step.

Consider the objection function,

$$J(\theta) = \sum_{s \in \mathcal{S}} \rho^{\pi_{\theta_{\text{old}}}} \sum_{a \in \mathcal{A}} \left( \pi_\theta(a|s) \hat{A}_{\theta_{\text{old}}}(s,a) \right),$$ (6)

where $\rho^\pi$ denotes the distribution over states following the policy $\pi$, $\theta^{\text{old}}$ is the policy parameters before the update, $\hat{A}$ denotes the estimated advantage function. Let the behaviour policy $\beta$ be $\pi_{\theta_{\text{old}}}$ and combines the advantage function, Eq. (6) can be rewritten as

$$J(\theta) = \mathbb{E}_{s \sim \rho^{\pi_{\text{old}}}, a \sim \pi^{\text{old}}} \left[ \frac{\pi_\theta(a|s)}{\pi^{\text{old}}(a|s)} \hat{A}_{\theta_{\text{old}}}(s,a) \right].$$ (7)

TRPO aims to maximize Eq. (7) by using policy gradient methods. Besides, TRPO constructs an extra constraint to enforce the parameter update not causing large variance, that is

$$\mathbb{E}_{s \sim \rho^{\pi_{\theta_{\text{old}}}}} \left[ D_{\text{KL}} \left( \pi_{\theta_{\text{old}}}(. \mid s) \| \pi_\theta(. \mid s) \right) \right] \leq \delta,$$ (8)

where $D_{\text{KL}}$ denotes the KL divergence of two probability distributions. By constraint (8), the old and new policy will not be large so that the training process is stabilized. However, although TRPO has beautiful theoretical formation, it is time-consuming to calculate the KL divergence in reality. Therefore, we consider using a more efficient algorithm called proximal policy optimization (PPO) Schulman et al. (2017) to simplify the calculation. PPO uses a clipped objection function, and it is given by

$$J^{\text{CLIP}}(\theta) = \mathbb{E}_{s \sim \rho^{\pi_{\text{old}}}, a \sim \pi^{\text{old}}} \left[ \min \left( \nu(\theta) \hat{A}_{\theta_{\text{old}}}(s,a), \text{clip}(\nu(\theta), 1-\epsilon, 1+\epsilon) \hat{A}_{\theta_{\text{old}}}(s,a) \right) \right],$$ (9)

where $A(\cdot)$ denotes the advantage function, $\nu(\theta) = \frac{\pi_\theta(a|s)}{\pi^{\text{old}}(a|s)}$ denotes the ratio between the old and target policy, and the function $\text{clip}(\nu(\theta), 1-\epsilon, 1+\epsilon)$ clips the ratio to be between $1-\epsilon$ and $1+\epsilon$. $\rho^\pi$ denotes the distribution over states following the policy $\pi$. The objective function of PPO takes the minimum one between the original value and the clipped one. As a result, we lose the motivation for increasing the policy update to extremes for better rewards. In implementation, to encourage the exploration, the objective function is given by

$$J^{\text{CLIP}'}(\theta) = \mathbb{E}_{s \sim \rho^{\pi_{\text{old}}}, a \sim \pi^{\text{old}}} \left[ J^{\text{CLIP}}(\theta) - c_1 \left( V_\theta(s) - V_{\text{target}} \right)^2 + c_2 H \left( s, \pi_\theta(\cdot) \right) \right],$$ (10)

where $c_1$ and $c_2$ are two hyperparameter constants, $H(\cdot)$ denotes the entropy function, and $V_{\text{target}}$ denote the sate value which can be calculated by using the sampled trajectories. This error term is generally added in Actor-Critic architecture. Besides PPO-CLIP algorithm used in this work, other RL algorithms like A3C Mnih et al. (2016), DDPG Lillicrap et al. (2015) and SAC Haarnoja et al. (2018) can also be used to construct our RGRL algorithm. The pseudocode of PPO-RGRL algorithm is presented in Algorithm 1.

## C  HYPERPARAMETER SPECIFICATIONS IN RGRL

The hyperparameters of PPO-Clip based RGRL algorithm are presented in detail in this section. In general, RL algorithm has more hyperparameters than supervised machine learning models. It generally includes the number of layers, the number of neurons of each layer with respect to actor and critic neural network, and the learning rate $\alpha$. Except for the general hyperparameters, particular RL parameters includes the total number of steps $M$, the number of steps per policy roll out $k_{\text{step}}$, the mini-batch size $B$, the number of epochs $K$ to update the policy, the discount rate $\gamma$, the surrogate clipping coefficient $\epsilon$, the entropy coefficient $c_2$, and the value loss coefficient $c_1$. Besides, to stabilize training, we use the gradient clipping strategy and set the maximum gradient norm allowed to be $g_{\text{max}}$. We also use the learning rate annealing technique, which aims to linearly decay the learning rate. The decay equation is given by

$$\alpha_k \leftarrow \left( 1 - \frac{k-1}{M} \right) \times \alpha_0,$$ (11)

where $\alpha_0$ is the initial learning rate. In reality, we find the algorithm performance is not sensitive to hyperparameters. All examples in this work share the same RL-specified hyperparameters. We

---

**Algorithm 1:** PPO-RGRL

---

**input** : Initial policy (actor) parameters $\theta_0$, initial value (critic) function parameters $\phi_0$; The maximum length (horizon) of each episode $T$.

**for** $k = 0, 1, 2, \cdots$ **do**

    Collect sampled trajectories $\mathcal{D}_k = \{\tau_i\}$ by running current policy $\pi_k = \pi(\theta_k)$ in the environment with $\tau_i = (s_i, a_i, r_k, s_{\rho_{i+1}})$.

    Compute the return (rewards-to-go) $\hat{G}_t$;

    Compute advantage function estimates, $\hat{A}_t(s_t, a_t) = Q^{\pi}(s_t, a_t) - V^{\pi}_{\phi_k}(s_t)$ based on the current value function $V^{\pi}_{\phi_k}$.

    Update the policy by maximizing the PPO-Clip surrogate,

$$\theta_{k+1} = \arg\max_{\theta} \frac{1}{|\mathcal{D}_k| T} \sum_{\tau \in \mathcal{D}_k} \sum_{t=0}^{T} J^{\text{CLIP}'}(\theta_k),$$

    via stochastic gradient ascent with Adam method Kingma & Ba (2014).

    Fit value function by regression on mean-squared error:

$$\phi_{k+1} = \arg\min_{\phi} \frac{1}{|\mathcal{D}_k| T} \sum_{\tau \in \mathcal{D}_k} \sum_{t=0}^{T} \left( V_{\phi}(s_t) - \hat{G}_t \right)^2,$$

    typically also via stochastic gradient descent algorithm such as Adam.

| Hyperparameter name | Value |
|:---:|:---:|
| $M$ | $50,000$ |
| $B$ | 64 or 128 |
| $K$ | 4 |
| $k_{\text{step}}$ | 512 |
| $\gamma$ | 0.99 |
| $\epsilon$ | 0.2 |
| $c_1$ | 0.5 |
| $c_2$ | 0.01 |
| $g_{\text{max}}$ | 0.5 |
| $\alpha_0$ | $4 \times 10^{-4}$ |

Table 1: The hyperparameters used in RGRL algorithm for all control examples. Fine-tuning these hyperparameters may further improve the training efficiency.

present these hyperparameters in Table 1. The RL-specified hyerparameters can be further optimzied to enhance the performance such as the sample efficiency.

Moreover, the hyperparameters of actor and critic neural networks are presented in the following. All actor and critic neural network contain three layers. The first layer maps the partial observation (neural representation of current quantum state) into the next hidden layers. The number of input neurons equals to the dimension of the neural representation $d$. The hidden layer contains 64 or 128 neurons and the last output layer contains only single neuron for citric network and $n_{\text{actions}}$ neurons for actor network. The activation function used in RGRL is $\text{Tanh}(\cdot)$ non-linear function. Note that in case $d = 32$, the number of neurons in hidden layer is set to be 64. In case $d = 96$, the number of neurons in hidden layer is set to be 128.

For different examples, the dimension of neural representation and the number of actions are also various. In all examples, we make use of discrete policy to control the parameters of quantum systems. Generally, to enhance the control efficiency, we make use of multi-discrete control policy to interact with the environment except for the XXZ model. It is quite intuitive since the parameters of quantum systems or Hamiltonian can be viewed independently from each other. For example, for Ising example, the number of actions is 18, i.e. $n_{\text{actions}} = 18$ as each parameter $J_i$ has three possible actions, namely $+1, -1$ and $0$. For XXZ model, the actions is set to be 8 as there are $n_{\text{actions}} = 8$

possible movements in the 2-dimensional parameter space except for the case of no movement. More specifically, each parameter can move towards left or right. Only one parameter moves or two parameters move simultaneously. For Kerr system, it is desirable to control the amplitude and phase of the light field. The amplitude and phase can be independently controlled and each has 3 possible movements. Therefore, there are $n_{\text{actions}} = 6$ possible movements in this environment. For quantum state retrodiction, the number of actions are also set to be $n_{\text{actions}} = 6$ as the gate parameters of initial quantum state control the amplitude and phase of one coherent state.

One more important hyperparameter is the discrete action step $\Delta a$ for each control example. The specific value is presented in Table 2. For Kerr model, the action step in each control loop is linearly

| Quantum Model | Discrete action step | Number of observables to measure |
|---|---|---|
| Random Ising model | $\Delta a = 0.05$ | $n_{\text{obs}} = 5$ |
| XXZ model | $\Delta a = 1$ | $n_{\text{obs}} = 50$ |
| Kerr model | $\Delta a = 0.3$ (linear decay) | $n_{\text{obs}} = 3$ |
| Generate output state of an unknown process | $\Delta a = (0.09, 0.06\pi)$ | $n_{\text{obs}} = 3$ |

Table 2: The action steps and number of observables to measure for each quantum system.

decayed following the same rule with Eq. (11). It is worthy noting that the discrete action step will determine the convergence of the proposed algorithm. A moderate action step is necessary to obtain a higher fidelity as the fidelity may be sensitive to control parameters. During the late stage of the training, the action step should be small enough to ensure the convergence with high fidelity. In general, all examples can make use of the strategy of linear decay strategy. However, it is not our focus in this work to optimize this hyperparameters to obtain the optimal performance.

## C.1 ADDITIONAL RESULTS ON CONTROL OF DISORDERED ISING MODEL GROUND STATE

We apply our RL algorithm for controlling the ground state of a disordered Ising model towards a target ground state. Specifically, suppose the quantum state under control is a 6-spin ground state of Hamiltonian

$$H_I := -\sum_{i=0}^{4} J_i \sigma_i^z \sigma_{i+1}^z - \sum_{j=0}^{5} \sigma_j^x, \tag{12}$$

where each $J_i \in (-1, 1)$ is an independent parameter. In this task, at each step, we must determine the change of each individual parameter, yielding a difficult high-dimensional control problem.

We suppose the quantum state under control is initialized either in the state $|+\rangle^{\otimes 6}$ that is the ground state when $J_i = 0$ for each $i$, or in a ground state corresponding to a randomly parameterized model in Eq. (12). We consider four control scenarios, each of which corresponds to a different target state: (1) a ground state in ferromagnetic phase corresponding to $J_i = 0.8$ for $0 \leq i \leq 5$, (2) a ground state in antiferromagnetic phase corresponding to $J_0 = J_2 = J_4 = 0.8$ and $J_1 = J_3 = J_5 = -0.8$, (3) a ground state corresponding to $J_0 = J_1 = J_2 = 0.8$ and $J_3 = J_4 = J_5 = -0.8$, and (4) a ground state corresponding to a randomly chosen Hamiltonian parameters. We randomly choose five 6-qubit Pauli bases out of $3^6$ possibilities and measure each single qubit, recording the outcome frequency distributions. At each control step, we perform the same set of quantum measurements. The measurement outcome statistics are fed into the neural network and then the RL algorithm outputs the actions on all six independent Hamiltonian parameters, where the action on each parameter is chosen out of the set $J_i \leftarrow \{J_i + 0.1, J_i, J_i - 0.1\}$.

Figure 8a illustrates the quantum fidelity between the controlled state after 50 control steps and the target state in four different control scenarios including both types of quantum initial states. The quantum fidelity achieved starting from $|0\rangle^{\otimes 6}$ is higher than those achieved starting from a randomly disordered initial state, indicating that controlling a general disordered state towards a target state is much more difficult. In Figure 8b, we show both the quantum fidelity and reward for the control scenarios (2) and (3) at every control step from the beginning to 100 steps.

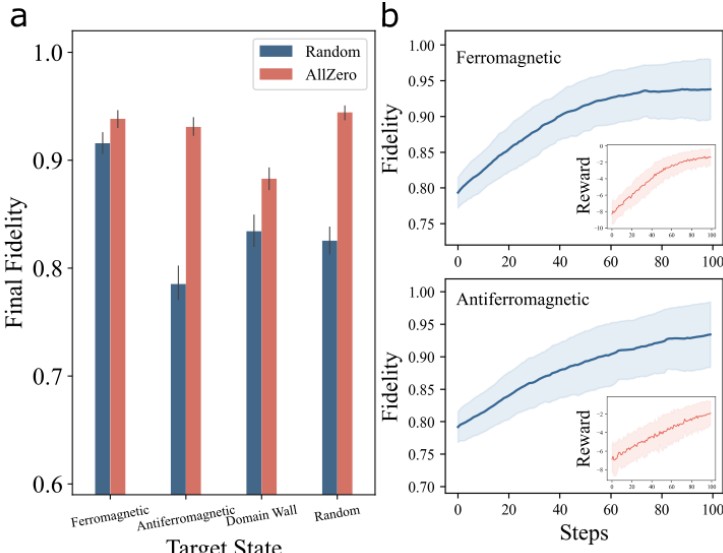

Figure 8: Control of Ising model ground states. Subfigure **a** shows the quantum fidelity between the controlled state and the corresponding target state in the four different control scenarios after 100 control steps averaged over 100 experiments. The vertical line at the top of each bar represents the confidence interval with confidence level 95%. Subfigure **b** shows the quantum fidelity of the controlled state in 100 control steps, together with the reward at every step.

## D CONTROLLING THE PREPARATION OF CV CAT STATES

Our RGRL algorithm can also be applied for controlling CV quantum states. Here, we focus on manipulating a superposition of two coherent states, known as a cat state $|\mathcal{C}_\alpha\rangle \propto |\alpha\rangle + |-\alpha\rangle$, where $|\alpha\rangle$ and $|-\alpha\rangle$ represent coherent states with amplitudes $\alpha \in \mathbb{C}$ and $-\alpha$, respectively. This cat state serves as an eigenstate of the Hamiltonian $H_{\text{Kerr}} := -\hat{a}^{\dagger\,2}\hat{a}^2 + \alpha^2\hat{a}^{\dagger\,2} + \alpha^{*\,2}\hat{a}^2$ Dykman (2012), with $\hat{a}$ and $\hat{a}^\dagger$ denoting annihilation and creation operators, respectively. Tuning the parameter $\alpha$ within the Hamiltonian $H_{\text{Kerr}}$ enables control the preparation of the cat state $|\mathcal{C}_\alpha\rangle$ Puri et al. (2017); Grimm et al. (2020), which is useful for quantum error correction Mirrahimi et al. (2014).

Here we consider four different control scenarios, each sharing the same target state $|\mathcal{C}^+_{0.5-1.8\text{i}}\rangle$, but starting from different initial states, as shown in Figure 9 (The initial state amplitudes are represented by black squares, and the target state amplitude is represented by a red star). We randomly choose three different quadratures $\left(e^{\text{i}\theta}\hat{a}^\dagger + e^{-\text{i}\theta}\hat{a}\right)/2$ associated with phases $\theta \in [0, \pi)$ and the homodyne measurements on these three quadratures form the set of measurements $\mathcal{S}$ at every control step. Each homodyne measurement outcome is binned into one of 100 possibilities. At each control step, the outcome frequency distributions are fed into the neural network, and our RGRL algorithm outputs one control action out of eight options $\alpha \leftarrow \{\alpha \pm \beta, \alpha \pm \beta\text{i}, \alpha \pm \beta \pm \beta\text{i}\}$, where $\beta$ denotes the size of the parameter change at each control step.

Figure 9**a** shows the quantum fidelity between the controlled state and the target state in four different scenarios after only 20 control steps. The figure includes both cases of ideal measurement statistics and measurement statistics with shot noise due to finite sampling. In the noiseless case, the quantum fidelity always exceeds 0.95 for each initial state. In the noisy case, we simulate the shot noise by adding Gaussian white noise with a variance of 0.1 to each frequency and renormalizing the overall frequency distribution. The resulting quantum fidelity is degraded by shot noise when the initial cat state has a large amplitude. Figure 9**b** shows the trajectories of controlled amplitudes in different control scenarios. Although the Wigner functions of the four initial states indicate significant differences, the final amplitude of the controlled state is close to that of the target state in all cases.

Now we demonstrate a different scenario in which we randomly select three different measurement quadratures at each control step, rather than keeping them consistent throughout the entire process

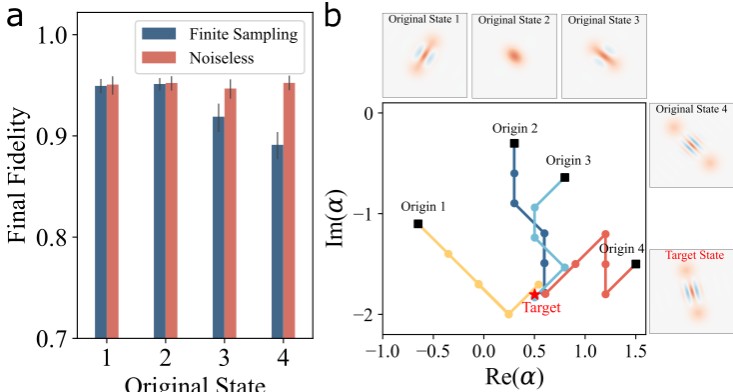

Figure 9: Control of preparing cat states. Subfigure **a** shows the quantum fidelity between the controlled state and the target cat state in four different scenarios after 20 control steps, averaged over 100 experiments. This includes both the noiseless case and the shot-noise case. Cases 1, 2, 3, and 4 correspond to four different control scenarios depicted in Subfigure **b**. The vertical line at the top of each bar represents the confidence interval with confidence level 95%. Subfigure **b** illustrates four trajectories of amplitude under quantum evolution in phase space, along with the Wigner functions of four different initial states and one target cat state.

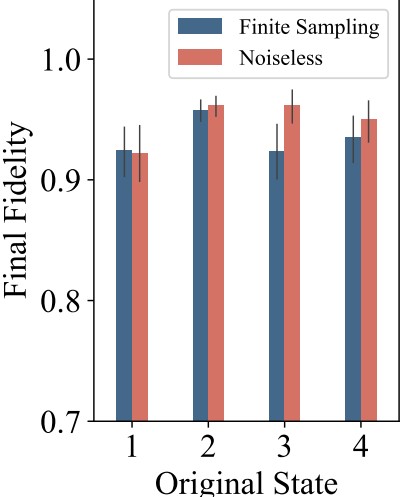

Figure 10: Control of preparing cat states in the scenario where measurement quadratures are randomly selected at each control step. The figure shows the quantum fidelity between the controlled state and the target cat state in four different scenarios after 20 control steps, averaged over 100 experiments.

of preparing the target cat state. We utilize the same four scenarios to test the performance of our proposed algorithm. Figure 10 shows the quantum fidelity between the controlled state and the target cat state in these four different scenarios after 20 control steps.

As shown in the results, the performance of our algorithm in this scenario surpasses its performance in the scenario where the measurement quadratures are fixed throughout the entire process. We believe this improvement is because the neural network collects more comprehensive information about the state when the measurement quadratures are varied. This variability likely enables the network to capture a richer set of state features, thereby enhancing the control precision and ultimately leading to higher quantum fidelity.

# E   LIMITATIONS

In this study, we focus on controlling the ground states of a Hamiltonian or the output state of an unknown quantum process. While these tasks are critical for quantum information processing, the types of quantum states currently under consideration remain limited to specific categories. In the future, it will be possible to generate a wider variety of quantum states using tailored quantum circuits, which can then be utilized to learn neural representations. Ultimately, our proposed RGRL algorithm can be applied to efficiently control these quantum states. To achieve this, we can train on a diverse set of quantum states generated by highly parameterized circuits, offering significant flexibility. A representation network, such as a large language model, could then be used to learn the structure of these quantum states. Looking ahead, large language models could play a crucial role in characterizing quantum states with sufficiently broad variability, enabling RL to efficiently manage both the dynamic evolution and computational behavior of these states.

