# OpenReview forum: "RGRL:  Quantum State Control via Representation-Guided Reinforcement Learning"
_ICLR.cc/2025/Conference — Submitted to ICLR 2025_

### Official Review · Reviewer_xcHQ · 2024-10-21

**Soundness:** 2
**Presentation:** 2
**Contribution:** 2
**Rating:** 5
**Confidence:** 3

**Summary:**

The authors develop a machine learning algorithm that characterises an unknown quantum state or unknown quantum operations by generating an abstract representation. The authors then show how reinforcement learning can be used to apply adequate control operations to realise a desired target state representation.  The control actions are determined by a neural network based on measurement data from a set of quantum measurements. This builds upon previous work  (Yan Zhu, Ya-Dong Wu, Ge Bai, Dong-Sheng Wang, Yuexuan Wang, and Giulio Chiribella. Flexible learning of quantum states with generative query neural networks. Nat. Commun., 13(1):6222, 2022.) which described a method of generating representations of quantum states but the novelty lies in the reinforcement learning algorithm.

This particular approach is applied to two systems of interest. First the authors show that they can effectively control many body ground states and realise desired phase transitions. Two different state representations; one for predicting measurement statistics and one for predicting mutual information are explicitly compared and it is shown that higher quality state representations yield better control efficiency. Lastly, the authors consider a control task where they prepare a target output state from a CV Kerr quantum gate.

**Strengths:**

This paper proposes an interesting extension of previous work in using an abstract representation of an unknown quantum state as a framework for performing "quantum control" and steering the system into a desired target representation. This seems a useful approach for tackling the control of black box systems where no information about the dynamics or the initial state is given.

The background on quantum states and measurements, as well representation network structure is described clearly and in a way such that non specialists can easily follow the authors exposition. The RGRL algorithm is also described clearly and Fig. 1 provides a straightforward description of the precise operations which are implemented.

The analysis of the physics of phase space transitions, as well as the complexity of the control task is also very thoroughly portrayed, this seems particularly appealing to readers with a physics background.

**Weaknesses:**

There are a number of issues with this paper which I would like to address.

Firstly, it seems as though it lacks the significance for it to be relevant to the wider audience of ICLR. The algorithm described in the paper seems interesting, but it is not clear that it is solving a broadly relevant and useful problem. The method is applicable particularly to systems where the initial states of the system are completely unknown, however no concrete examples of important and relevant physical systems which exhibit this behaviour are shown or described. Similarly, in the case of the CV Kerr gate, it is not clear what physically motivates the experimental setup: where a "target output state can be obtained by applying the unknown quantum process to a certain input state". It would be helpful to frame this more clearly with explicit references.

Secondly, It is not clear to what extent the algorithm is practically useful when sampling from a real quantum device, since "M" copies of the same quantum state need to be prepared (what about the no cloning theorem?). Claims in the paper are made that the number of measurement samples is "small". It would be useful to have a quantitative comparison of sample efficiency compared to other methods, and to discuss how the approach handles the constraints of real quantum devices, including preparation and measurement times. For example, the authors could look at the following reference: Irtaza Khalid, Carrie A. Weidner, Edmond A. Jonckheere, Sophie G. Schirmer, and Frank C. Langbein Phys. Rev. Research 5, 043002 which constructs a sample efficient RL algorithm for quantum control.

Thirdly, there are no benchmarks or comparisons drawn to previous work in the quantum control literature. From the presentation in the paper, it is not clear whether the method outperforms previous methods or approaches in any relevant quantum control tasks. Moreover, the authors claim their algorithm is "scaleable", but it is not clear what makes the method scaleable and there are no explicit benchmarks. There are some alternatives in the literature which could be compared and contrasted; for example given uncertainty about the initial state, one can incorporate this into the design of a particular quantum control sequence which does not require a "black-box treatment" as shown in (Frank Schäfer et al 2020 Mach. Learn.: Sci. Technol. 1 035009) where the authors claim that "Despite the uncertainty in the initial states, we managed to reach fidelities larger than 99.9% in the preparation of a GHZ state in a chain of M qubits." It is also unclear whether one would opt for a "black-box" method when the initial state is unknown in a real physical system, as often a more efficient approach in real quantum experiments would be to bring the quantum state into a known quantum state to then perform a control operation (e.g. optical pumping in atoms or transmon reset).

Lastly, the manuscript is not always clear and some of the experimental results are inaccessible to a non-specialist audience and I suspect the broad audience of ICLR. The Kerr gate is mentioned but no citation is given for further details and it is not clear why this is a particularly important problem to tackle which should be explained. Moreover, the control of many body quantum states is not sufficiently contextualised or motivated. The in-depth discussion of different types of phase transitions as well as Figs 3/6 showing "representation space" are not clear (no axes labels). The "t-SNE algorithm" is also left unexplained with no citation and it would be good to better explain this for a reader with an ML background or cut down on specific details on the phase transitions. Some more explicit references need to be made to the Appendix sections to clearly guide the reader for further background/implementation details.

**Questions:**

In the introduction you mention that "a few works explored the possibility of directly using measurement outcomes for reward calculation Reuer et al. (2023); Borah et al. (2021); Sivak et al. (2022), but scalability still remains a challenge." Is your algorithm more scaleable than the alternatives and why? You claim that "This approach significantly reduces the reliance on precise initial state information, making the algorithm scaleable and applicable to larger quantum systems". How does the algorithm scale with quantum system size and what makes it scaleable?

How many measurements are needed and how does this scale with representation quality? How does this compare with other black box approaches in terms of the number of required samples?  Perhaps this is also an interesting Reference to consider: Yuval Baum, Mirko Amico, Sean Howell, Michael Hush, Maggie Liuzzi, Pranav Mundada, Thomas Merkh, Andre R.R. Carvalho, and Michael J. Biercuk PRX Quantum 2, 040324.

What particularly physical scenarios/experiments motivate the study of a system with an unknown initial state or completely unknown quantum operation?

Why is uncertainty in the initial state a "major challenge in quantum state control"?

Is this method more applicable to unknown initial states or unknown quantum dynamics?

How does this method compare (for instance in terms of achieving higher fidelity control solutions, more robust solutions etc..) to previous methods in quantum control?

What is your ML contribution? It seems like you are applying a standard algorithm (PPO) to a particular way of representing quantum states with limited measurements, or is there more to the algorithm that may have escaped my attention?

You say: "we use the distance between quantum state representations as a proxy for quantum fidelity", but then explicitly refer to the "quantum fidelity" at numerous occasions when describing experimental results. Do you mean the quantum fidelity? Or can the relationship between the representation distance and quantum fidelity be explicated more explicitly.

---

### Official Review · Reviewer_776H · 2024-11-03

**Soundness:** 3
**Presentation:** 3
**Contribution:** 2
**Rating:** 3
**Confidence:** 3

**Summary:**

In the control of quantum systems, a common scenario is the task of preparing a particular quantum state. In simulations, one would be able to adjust controls based on of the current (simulated) quantum state, including the initial state, but that is not possible in a real experiment and one only has access to the measurement statistics. This article proposed a new reinforcement learning method termed RGRL. It is built on two previous works [Zhu et al (2022) and Wu et al. (2023b)] which introduced networks that learn to translate measurements into a representation of a quantum states. In every control step, one would translate measurement statistics into such a representation and decide on the next step base on that information. The representation networks are trained before controls are trained.

  The authors demonstrate the application of the new method for two examples: Preparing target states in the XXZ model as a many-body example, and in a continuous variable system as a single-system example.

**Strengths:**

- The method and its application is presented clearly, the work is of high quality technically.
- The problem of state preparation is relevant, the examples are well-chosen, and the new method achieves its task of state preparation.
- The presented algorithm seems plausible, could be promising, and seems like a natural application of the [Zhu et al (2022) and Wu et al. (2023b)] articles.

**Weaknesses:**

While the algorithm does not require pre-existing knowledge of the quantum system’s dynamics, it does require knowledge of the possible system states to train the representation networks it is based on. Direct end-to-end learning based on the measurement statistics would be truly independent of prior knowledge and the comparative advantage of the method RGRL presented here is unclear to me. To demonstrate any such advantage, the RGRL method would have to be benchmarked against prior work, for example the articles cited in the introduction. Such benchmarking would significantly strengthen the article, going beyond the more exploratory demonstration of RGRL.

**Questions:**

Can you demonstrate an advantage over previous methods or more direct approaches such as end-to-end learning without the intermediate step of the pre-trained representation networks?

---

### Official Review · Reviewer_H4g3 · 2024-11-03

**Soundness:** 2
**Presentation:** 2
**Contribution:** 1
**Rating:** 3
**Confidence:** 4

**Summary:**

The paper proposes improved quantum state control via taking small initial samples of the system to build an improved prior RL model and thus learn actions more appropriately. Contribution is a RL algorithm to steer an uncharacterized quantum state towards a target state using only measurement statistics. The algorithm uses a representation network to estimate state representations and their similarity to the target state. With the trained representations, the model can also be used to incrementally search for inputs that produce a certain target output of given unknown quantum process. The algorithm is tested on control of many-body ground states across phase transitions, like trivial or topological symmetry broken phases.

**Strengths:**

Well written, clear and concise. Not to much jargon, nicely understandable even from an outside perspective. I also found the figures conceptually nicely understandable with the differently colored phase regions. The application seems to be works well, empirically.

**Weaknesses:**

Weaknesses:

As major weaknesses the following issues stood out to me:

- Duplicate explanation of the environment, reward and the representation input in 3.1 and 3.3. Those two sections should be consolidated. Also the reward in 3.1 is introduced first by the simple negative euclidian distance, but the EQ uses a different norm ||.||(_2?) and divides by d. This should be more consistent.
- Its not clear if the architectures in Figure 2 are part of the contribution? As I understand, Figure 2 is simply an illustration of representation network and decoders of Wu et al. 2023b?
- Figures 3 d/e, 6 and 7 are way too small to read, even with full digital zoom. Most of the explanation focus on Figure 3, with Figures 4-6 only barely mentioned in L350-354. Even with the caption, I am not sure I understand what is meant to be conveyed with Figure 7.
- Novelty in this work is rather lacking, in my opinion. This is a simple application of RL to a discrete control problem with a very simple reward function. The representation network is an interesting inclusion, but similarly also only an application of existing work. The choice of designs (decoder architecture, RL algorithm choice, hyperparameters etc.) are not motivated, although part of this information can be found in the appendix. (Still papers should be self-contained without the appendix.)
- Finally there is no comparison to other approaches included in this paper and as such significance is hard to judge. Without context, or at least an ablation study of different choices for the chosen approach the contribution of this paper is very light.

---

Minor Notes:

- Figure 1 does not really present any valuable insights than one line of text would not also have explained.
- L154, 158 i.e.[,]
- L 157 non[]Gaussian
- L 182 unclosed bracket and whitespace ([]from a finite set …
- The notation of bold-r for representations and default r for reward could be chosen better and are confusing at times.
- The notation of maximizing the average cumulative reward is worded badly. In any RL setting, the RL agents learns to optimize a policy that maximizes the reward / return of each episode. Improvement of average return over time is simply a side-effect of this.
- Inconsistent use of Fig. and Figure. for references.
- L 327, 329 [Ref.] {citation} ?
- The color palette in Fig.3 d/e could be better chosen and the figures should be bigger. The light blue/beige colors and the trajectory, at that small scale, are very hard to see.
- L 371 overloads the variable “r” for the third time.

**Questions:**

- L 346 claims: “Thus, the optimal path in phase space does not correspond to the shortest trajectory in the representation space, implying that the control task we consider is highly nontrivial.” However, t-sne simply arranges the learned representations in low-dimensional space, and unless the pretraining of the representation network included some form of latent-space structuring, the projections will always be of chaotic nature. As such I find it difficult to reason about the complexity of this task by the t-sne control trajectory, could you please elaborate a bit more on this?
- Is there a reason as to why Figure 3 only shows 30 steps? Does the algorithm perform better / finds the target state eventually, or are the 30 steps a physical hard-constraint? (Figures 7 are plotted up to 55 control steps?)
- In Figure 3a, the no-label case for Tr→SB is very much an outlier, performance wise. Is there a reason for this?

---

### Official Review · Reviewer_D3q3 · 2024-11-04

**Soundness:** 3
**Presentation:** 2
**Contribution:** 2
**Rating:** 6
**Confidence:** 4

**Summary:**

The paper provides an approach for the extraction and representation of states from a quantum system, improving the reward calculation for training a reinforcement learning policy in quantum control tasks.

**Strengths:**

The paper is theoretically well-elaborated and provides a sound approach for improving state extraction in RL for quantum control. It is generally well-written and provides solid connections to the background in quantum physics. Overall, the paper presents a well-motivated method to improve exploration for reinforcement learning in quantum control by providing a smooth reward signal.

**Weaknesses:**

Despite being motivated for NISQ Devices, the paper does not provide evaluations regarding those. While the paper provides solid connections to the background in quantum physics, I am missing such connections to the background in reinforcement learning, especially focusing on integrating the extracted state and reward signal into existing frameworks for RL in quantum control and quantum circuit design. Generally, I feel like the claimed contribution could be clarified, as the paper does not seem to provide an RL algorithm but rather a method for improved state representation, which is still an important contribution when connected to existing frameworks for RL in this domain. Regarding the empirical results, I am missing details on the training process, comparisons to considerable baselines, and, in general, quantitative results. Also, the results shown in Fig. 3 could be better described in the text. Finally, the potential limitations of the approach should be discussed.

Minor comments:

- The stochastic policy should be defined as $\pi: D \times A\mapsto[0,1]$
- r seems overloaded, consider changing representation or reward to avoid confusion
- A common notation t for a timestep should be used
- Citation formatting could be improved (e.g., using \citet for in-text citations).
- The placement of figures (e.g., Fig.3) could be improved to ease readability

**Questions:**

How well does the proposed approach generalize to recalibrations? Is overfitting the training device a potential issue to be considered?

Could the proposed method be used as an extension to existing QRL frameworks (e.g., [1-4])?

[1] van der Linde et al., 'qgym: A Gym for Training and Benchmarking RL-Based Quantum Compilation', 2023.
[2] Altmann et al., 'Challenges for Reinforcement Learning in Quantum Circuit Design', 2023.
[3] Kölle et al., 'A Reinforcement Learning Environment for Directed Quantum Circuit Synthesis', 2024.
[4] Rietsch et al., 'Unitary Synthesis of Clifford+T Circuits with Reinforcement Learning', 2024.

---

### Official Review · Reviewer_65aL · 2024-11-04

**Soundness:** 2
**Presentation:** 3
**Contribution:** 1
**Rating:** 3
**Confidence:** 4

**Summary:**

The paper tackles the well-known issue of quantum state control, i.e., steering a quantum system towards a target quantum state. To this end, the authors run reinforcement learning with the addition of a learned state representation encoded in a representation network. They show that their approach can steer quantum systems in a meaningful way.

**Strengths:**

The paper tackles a very relevant problem.

I believe the proposed approach has great merit. Training a dedicated representation network sounds like a good idea (as it is also well motivated in the paper) and might have a big impact on various quantum applications.

The paper explains the approach well and features several amazing visualizations.

**Weaknesses:**

The main problem is that the presented approach is not sufficiently analyzed. Several design decisions in the algorithm and in the study are not really discussed and not analyzed. There is no ablation study or a comparison to other candidate approaches. There is no application of the approach to other similar issues.

The training behavior and training properties of the presented approach are neither shown nor discussed. Thus, this paper's contribution to an AI community is unclear.

The setting that reinforcement learning operates in is not formalized in a standard way and thus hard to follow. Giving a standard MDP-style definition would help here.

Several typos persist. Most importantly, all references are formatted incorrectly.

**Questions:**

None.

---

### Meta-Review · Area_Chair_UaHX · 2024-12-11

**Metareview:**

This paper introduces an approach for quantum state control using reinforcement learning guided by representation learning. While the problem addressed is relevant, the paper suffers from significant shortcomings that limit its impact and clarity. Key concerns include the lack of empirical rigor, as there are no benchmarks or ablation studies to evaluate the proposed method against established baselines. The novelty of the approach is also limited, with the representation network primarily applying existing techniques without clear improvements over direct end-to-end learning methods. Furthermore, the paper does not adequately address practical challenges, such as scalability and applicability to real quantum systems with physical constraints like sampling efficiency. The presentation, while clear in parts, is inconsistent, with several figures and concepts underexplained, making it difficult for a broad audience to assess the results or significance. These limitations, combined with the lack of contextualization within the reinforcement learning and quantum control literature, justify a decision of Reject.

**Additional Comments On Reviewer Discussion:**

N/A

---

### Decision · Program_Chairs · 2025-01-22

Reject